# Evaluation of Collagen and Elastin Content in Skin of Multiparous Minks Receiving Feed Contaminated with Deoxynivalenol (DON, vomitoxin) with or without Bentonite Supplementation

**DOI:** 10.3390/ani9121081

**Published:** 2019-12-04

**Authors:** Iwona Taszkun, Ewa Tomaszewska, Piotr Dobrowolski, Andrzej Żmuda, Wiesław Sitkowski, Siemowit Muszyński

**Affiliations:** 1Sub-Department of Clinical Diagnostics and Veterinary Dermatology, Department and Clinic of Animal Internal Diseases, Faculty of Veterinary Medicine, University of Life Sciences in Lublin, Głęboka St. 30, 20-612 Lublin, Poland; wieslaw.sitkowski@up.lublin.pl; 2Department of Animal Physiology, Faculty of Veterinary Medicine, University of Life Sciences in Lublin, Akademicka St. 12, 20-950 Lublin, Poland; ewaRST@interia.pl; 3Department of Functional Anatomy and Cytobiology, Maria Curie-Sklodowska University, Akademicka St. 19, 20-033 Lublin, Poland; piotr.dobrowolski@umcs.lublin.pl; 4Department of Epizootiology and Clinic of Infectious Diseases, Faculty of Veterinary Medicine, University of Life Sciences in Lublin, Głęboka St. 30, 20-612 Lublin, Poland; andrzej.zmuda@up.lublin.pl; 5Department of Biophysics, Faculty of Environmental Biology, University of Life Sciences in Lublin, Akademicka St. 13, 20-950 Lublin, Poland; siemowit.muszynski@up.lublin.pl

**Keywords:** minks, skin, deoxynivalenol, bentonite

## Abstract

**Simple Summary:**

The presence of mycotoxins in products intended for consumption is harmful to the health of both people and animals. One of the most abundant mycotoxins in mink’s feed, often contaminating cereal grains, is a mycotoxin produced by the fungi *Fusarium* spp. deoxynivalenol (DON). The aim of the study was to investigate whether and how the long-term supply of this mycotoxin in feed influences the skin of adult female minks. An additional objective was to assess the effects of the bentonite additive to feed contaminated by DON, which has the ability to reduce the impact of mycotoxins. The scrapes of the skin were collected from animals after euthanasia and before pelting. After preparing histological slides, samples were examined microscopically. The thickness of the epidermis and dermis was investigated and the presence of elastin and collagen. These parameters determine the quality of the fur skins and economic aspect of this animal husbandry. The results showed that DON causes a decrease in the presence of total collagen and absence of immature collagen, thus reducing the elasticity and flexibility of the skin. The addition the bentonite to feed stimulates the production of collagen, restoring the proper relationship between the tested parameters in mink’s skin.

**Abstract:**

Deoxynivalenol (DON, vomitoxin) is considered one of the most dangerous mycotoxins contaminating cereal products for food and feed. One of the protective methods against the adverse effect of DON on mink health is to use a component such as bentonite as a feed supplement to allow toxins absorption. The aim of this study was to determine the effect of DON, administered alone or with bentonite, on the histological structure of the skin and the content of collagen and elastin. A multiparous minks from control group (not exposed to DON) and a study groups receiving fed with DON-containing wheat for seven months: I: at a concentration of 1.1 mg/kg of feed, II: at a concentration of 3.7 mg/kg, III: DON at a concentration of 3.7 mg/kg and bentonite at a concentration of 0.5 kg/1000 kg of feed (0.05%) and IV: DON at a concentration of 3.7 mg/kg and bentonite at a concentration 2 kg/1000 kg (0.2%). After performing euthanasia and before pelting, skin samples of 2 cm in diameter were drawn from the multiparous minks from the lateral surface of the right anterior limb. Our obtained results clearly indicate that DON administered for a period of seven months at a dose of 1.1 mg/kg significantly changes the thickness of skin of a multiparous mink. It causes an increase in the percentage of elastin from 5.9% to 9.4% and a decrease in immature collagen, which results in a change in the collagen/elastin ratio from 10/1 to 5/1. A dose of 3.7 mg/kg DON in feed without or with 0.05% bentonite causes the absence of immature collagen in the dermis, but the addition of 0.2% bentonite in the feed reveals the presence of immature collagen and increase the percentage of the elastin.

## 1. Introduction

Fur condition and skin size are two of the most important traits that determine pelt quality [1]. Anatomical structure of the skin living minks determines its durability, flexibility, and susceptibility to burst, which is important in the process of fleshing and tanning [2]. The correct structure of the epidermis, dermis, and subcutaneous tissue allows for stretching and forming in all directions that ensures high skin tear resistance during technological processing. The hydration and moisturizing of the epidermal cells affect the quality of hair and a proper keratosis of epidermis process [3,4]. Main constituents of the dermis, collagen, and elastin ensure elasticity and flexibility. One of the factors that influences the quality of mink skin is adequate nutrition. Component of the mink’s feed, which often contaminate cereal grains, is a trichothecene mycotoxin, mainly produced by fungi of the genus *Fusarium* spp. deoxynivalenol (DON, vomitoxin) [5,6]. DON, due to its common occurrence and strong toxicity, is considered one of the most dangerous mycotoxins contaminating cereal products for food and feed [7,8,9,10,11]. A dose of DON over 50 mg/kg body weight is considered to be the LD_50_ toxic dose [12]. According to the recommendations of the European Union, the DON content must not exceed 1.75 mg/kg in wheat grain and 0.75 mg/kg in flour and cereal products for human [5,13]. In livestock animals, feed DON contamination generates many clinical effects, such as anorexia, emesis, diarrhea, anemia, hemorrhage, weight loss, nervous disorders, skin toxicity, and bone marrow damage [14,15], but the sensitivity of different species of animals to DON varies. Ruminants and poultry tolerate up to 20 mg/kg of DON in feed, whereas 1–2 mg/kg causes toxicity in pigs [5,14,15,16]. One of the protective methods against the adverse effect of DON on animal health is to use as a feed supplement to allow toxins absorption, such as bentonite [17]. Bentonite, a natural and biologically active soil component, has been registered under the European Commission Implementing Regulation No. 1060/2013 as a technological feed additive without any restriction [18]. Its application involves inhibition of the growth of toxin-producing fungi in feed, yet this effect is dependent on bentonite type (smectite content) [17]. It was found that sodium bentonite, supplemented at the dose of 1%, was found to reduce fungus number in farmed minks’ feed [19] without adversely affecting animal health, as indicated by blood parameters [20]. 

The aim of this study was to determine the effect of DON, administered alone or with bentonite supplementation, which eliminates mycotoxicity from mink diet, on the histological structure of the skin and the content of collagen and elastin. 

## 2. Materials and Methods

The experimental procedures used throughout this study were approved by the Local Ethics Committee on Animal Experimentation of University of Life Sciences in Lublin, Poland (Resolution No. 64/2015).

### 2.1. Animals and Experimental Design

The location selected for the in southern Poland used for breeding purposes. Prior to research we chose 60 clinically healthy, dark brown type, multiparous American minks (*Neovison vison*). The individuals were kept isolated from each other in separate cages under common farming conditions and natural photoperiod routine, were given access to drinking water and underwent daily well-balanced, regular diet. The mating with regards to female individuals took place in March. Subsequent to mating, two groups of minks were randomly formed: a control group, C (n = 15; not exposed to DON) and a study groups fed with DON-containing wheat (each group n = 15): I: fed with DON-containing wheat at a concentration of 1.1 mg/kg, II: fed with DON-added wheat at a concentration of 3.7 mg/kg, III: fed with DON-containing wheat at a concentration of 3.7 mg/kg and bentonite at a concentration of 0.5 kg/1000 kg (0.05%) and IV: fed with DON-containing wheat at a concentration of 3.7 mg/kg and bentonite at a concentration 2 kg/1000 kg (0.2%) (Table 1).

DON groups were given contaminated feed starting first day following mating and throughout the gestation (circa 46th day), lactation, and reaching pelt collection period. To ensure normal pregnancy and to maintain unchanged feed intake in mink dams, the concentration of DON was selected based on available literature [21]. Wheat free of mycotoxins was mixed with wheat naturally contaminated with DON, which level were last determined by HPLC (high-performance liquid chromatography) according to the ISO/IEC17025 standard [11,22]. The DON content in given feed was also verified by HPLC analyses. As a detoxifier a bentonite (Mycofix, Biomin, Poland) at an amount of 0.5 and 2 kg/1000 kg of was used. The doses were selected on the basis of EFSA (European Food Safety Authority) recommendations and working levels recommended by the producer [18,23]. Following delivery (May, June), lactating dams were kept with their offspring until weaning (September, approximately after two months). Carbon monoxide inhalation was used as a method for euthanasia prior to pelt collection in compliance with farming procedures and Polish domestic laws [24,25]. The average euthanized multiparous mink body weight was within the range of 1477 g to 1952 g, with the reduced body weight observed in the group treated only with 3.3 mg/kg of DON (group II) but there were no visible signs of deterioration. Shortly after performing euthanasia and before pelting, with the use of a surgical blade, skin samples of 2 cm in diameter were drawn from the multiparous minks from the lateral surface of the right anterior limb. Prior to sampling, no stretching of the minks’ skin was performed.

### 2.2. Histomophometrical Analysis of Skin

With no prior stretching applied to skin samples, they were laid out flat in standard histological cassettes in a manner that prevented them from forming contact points with cassette’s walls. Subsequently, they were put in 4% solution of buffered formaldehyde (pH 7.0) for a 24 h period [26]. Afterwards, all samples were dehydrated and cleared in Ottix Plus and Ottix Shaper (DiaPath, Martinengo, Italy) solvent substitutes, respectively, and subjected to paraffin saturation, carried out with a tissue processor (STP 120, Thermo Scientific, Waltham, MA, USA). With the use of modular embedding device EC 350 (Especialidades Médicas Myr S.L., Tarragona, Spain), samples were fixed in paraffin. The samples were then sectioned and underwent histological evaluation as described in a previous study [27], forming 20 cross-sections (with 10 mm gap after each five-slice section). From each skin sample, with the use of microtome (Microm HM 360, Microm, Walldorf, Germany), 4 μm-thick slices were cut out and each of the five semi-serial cuts was put on separate microscopic slides. In forthcoming analysis, two from each five slices of each slide were subjected to evaluation. In order to differentiate between particular skin structures (epidermis, dermis, and subcutaneous tissue), we used Masson’s method of trichrome staining. To differentiate between types of collagen fibers found in connective tissue, picrosirius red staining method was employed. Thinner fibers (type III or immature collagen) were stained green, while thicker collagen fibers (type I or mature collagen) are stained red [28]. Verhoeff-van Gieson Stain (VVG) was used for elastic fibers differentiation and analysis, in accordance with the published protocol [29]. The stain was selected owing to its applicability in determining emphysema-specific elastic tissue atrophy as well as thinning and loss of elastic fibers.

Analysis was conducted using histological specimen microscopic images. The samples were analyzed with the use of a microscope at following magnification values: ×200, ×400. and ×1000 (Olympus BX63, Olympus, Tokyo, Japan) as well as software for graphical analysis (Olympus CellSens 1.5, Olympus, Tokyo, Japan).

In each histological sample, the thickness of both dermis and epidermis, given in µm, were evaluated 100 times, providing the mean value for further assessment. The measurement of epidermis’ thickness was taken from the *stratum basale* to the *stratum corneum*. In the case of dermis, it was taken from the subcutaneous fat and hair bulb layer to the basal layer.

At magnification ×200, the PSR-stained sections were evaluated with the use of filters adapted to provide circularly polarized lighting. The filters were aligned to make the background as dark as possible in the field of view i.e., the filters were ‘crossed’ as Tomaszewska et al. [26] describe it. Obtained microscopic images were analyzed with the use of ImageJ graphic analysis software (v. 1.51, National Institutes of Health, Bethesda, MD, USA). A color threshold tool was used to calculate the area of red (thick, more mature) and green (thinner, immature) collagen fibres in selected image sections by pixel counting method. As for fur shaft cortex, due to its birefringence and changes in light polarization, only sections deprived visible fur hair were analyzed [30].

Automated analyzes of elastic fibers were conducted on microscopic images of sections stained by the Verhoeff-van Gieson technique, using ×400 magnification under the light microscope in bright field mode. An ImageJ analysis software color threshold tool was used to select a range of dark brown to black elastic fibers from the cross section of the dermis images. After the abovementioned thresholding, the region of chosen fibers was counted on binary images by pixel counting method. As a proportion of dermis connective tissue, the quantity of elastic fibers was provided.

### 2.3. Statistical Analysis

All statistical analyses were carried out with the use of Statistica 12 software (StatSoft, Inc., Tulsa, OK, USA). Results are expressed as mean values (n = 15) ± SD (standard deviation). In order to detect any significant difference between the groups, one-way analysis of variance (ANOVA) followed by Tukey’s multiple comparison test were carried out. For all comparisons, the probability level of *p* < 0.05 and *p* < 0.01 were considered statistically significant. Correlations were analyzed using the Spearman’s rank correlation test. 

## 3. Results

The measured values of epidermis and dermis thickness are presented in Table 2. The thickness of epidermis was significantly increased in two groups exposed to 3.7 mg/kg of DON, namely group II and III, when compared to the control group (*p* < 0.01 and *p* < 0.05, respectively) and when compared to the group I, exposed to 1.1 mg/kg of DON (*p* < 0.05 for both groups). The 0.2% supplementation of bentonite (group IV) restored the normal thickness of epidermis. The presence of DON, irrespective of its dose or bentonite supplementation, caused a significant increase in the thickness of the dermis, at least two-fold, when compared to the control group (*p* < 0.01). However, the dermis thickness in group supplemented with 0.2% of bentonite (group IV) was significantly reduced when compared to the group deprived bentonite supplementation (group II, *p* < 0.05) and the group exposed to 3.7 mg/kg of DON and supplemented 0.05% of bentonite (group III, *p* < 0.01).

In control C group, the percentage of elastin in the dermis equals 5.9% and the collagen to elastin ratio was 10.13 (Figure 1 and Table 3). The DON contamination increased the percentage of elastin in all groups except group II (*p* < 0.01, *p* < 0.05, *p* < 0.01, for group I, group III, and group IV, respectively), while the decrease of collagen/elastin ratio was observed on all groups exposed to DON when compared to the control C group (*p* < 0.05 for all groups). However, the highest content of elastin and the lowest collagen to elastin ratio was observed in skin of minks from group IV exposed to 3.3 mg/kg of DON and supplemented with 0.2% of bentonite, as these values differed significantly form values observed in other DON-exposed groups. When compared two groups exposed to different doses of DON without bentonite supplementation (1.1 mg/kg in group I and 3.7 mg/kg group II) statistically significant differences in elastin content and collagen/elastin ratio are also observed (*p* < 0.05).

For the control C group, immature collagen accounted for 16.3% of mature collagen (collagen ratio 0.163) present in the skin (Figure 2 and Table 4). DON administered at a dietary dose of 1.1 mg/kg (group I) resulted in a decrease of the collagen fibers content in the skin (*p* < 0.05). In two groups exposed to DON a dose of 3.7 mg/kg, deprived bentonite supplementation (group I) and supplemented with bentonite at the dose of 0.05% (group III), immature collagen fibres (type III) were not observed (Figure 2). However, the addition of bentonite at a dose of 0.2% (group IV) restores the production of type III collagen, however its content was still significantly decreased when compared both to control C and group II (*p* < 0.01 for both).

Statistical analysis of the results revealed a high negative correlation between collagen and elastin content in all groups of animals, ranging from −0.678 in group II to −0.954 in group III (Table 5).

## 4. Discussion

However, literature does not offer much data on the quality assessment of minks’ skin. As indicated by the published data from Poland, the thickness of epidermis in healthy minks is on average at 6.4 µm, while for dermis this value is at 634.7 µm [2]. In our studies, the mean value of epidermal thickness in healthy minks was at 17.62 μm while with regards to dermis at 570 μm. Skin thickness depends on collagen and elastin fibers as well as cellular substances and fluid content. In dogs, increased fluid content lead to a greater skin thickness ranging 13–21% in different regions of the body [31]. In our study, the thickness of the skin increased by approximately 250%. The thickness of the epidermis and dermis was measured in histological skin preparations taken from the lateral surface of the right anterior limb from healthy minks which were naturally fed with DON-contaminated feed. This allowed the effects of the different levels of DON on the quality of the animal skin and the effectiveness of bentonite in eliminating the toxic effects of DON. Deoxynivalenol is one of the most prevalent mycotoxins produced by *Fusarium* spp. This trichothecene is regularly detected in food, among others, in wheat. The mechanism of toxic effects of DON is not conclusively explained. As many authors point out, the toxic effects of DON may be the result of dysregulation of immunological and/or neuroendocrine function [14,32]. The cellular mechanism of DON’s toxicity is explained by its effect on synthesis of protein and nucleonic acid. DON blocking the ribosomal active and inhibiting protein synthesis. DON enters cell and binds to active ribosomes which transduce a signal to RNA-activated protein kinase (PKR) and hemoitopoeitic cell kinase (Hck). Subsequent phosphorylation of protein kinases drives cells apoptosis and immunotoxic effects [7,14,32]. 

In normal skin of adult people, collagen fibers account for 90% of all dermal fiber (elastin fibres are not more than 10%) [33,34], thus the collagen/elastin ratio is 10:1. The same ratio is observed in healthy minks from the control group not exposed to DON, as collagen to elastin ratio is 10:1. We observed that DON causes an increase in the percentage of elastin. Moreover, at the dose of DON 3.7 mg/kg in feed, with or without the addition of 0.05% bentonite, the immature collagen is not identified in the dermis. This resulted in significant reduction of collagen to elastin ratio. Collagen present in skin tissue is mainly collagen type I: 75%, type III: 22%, and type V: 3%, so the collagen type I to type III ratio is about 2.5 [33,34]. In our previous works, in which we examined the effect of DON and bentonite on bone mechanical properties in minks, we suggested that malnutrition resulting from the presence of mycotoxicosis in the diet could impaired synthesis of collagen type I [35,36]. Moreover, the effect of mycotoxins (DON, Ochratoxin A) on collagen synthesis was previously observed in studies performed cell lines, showing that mycotoxins can influence the expression of fibrous collagens, including collagen type III [37,38]. Our present study confirms these previous assumptions and results of in vivo studies. What is more, we showed that the addition of 0.2% bentonite shows a positive effect on collagen synthesis, as it reveals the presence of immature collagen. This result increases the rationale for using bentonite on mink farms [18,19,20,39].

The results obtained by show that DON administered for a period of seven months significantly changes the thickness of skin of a multiparous mink. When explaining the changes in the mink skin through the mechanism of DON toxic effect described above, it can be assumed that the thickness of the epidermis grows due to the increase in the thickness of the *stratum corneum*, hydration disorders and swelling of the cells of the living layers of the epidermis (*stratum basale*, *spinosum* and *granulosum*). This is the stage of cell death. The mechanism of action of trichothecenes is not yet completely explained. It is generally known, that in a microscope image apoptosis, in contrast to necrosis, occurs with cell shrinkage by the loss of water. Apoptosis does not cause inflammation and affects individual cells. Typical apoptotic changes are observed during physiological aging of the skin. The skin becomes thin and it loses flexibility and elasticity due to water loss and loss of protein structural elements (collagen, elastin). Therefore, the results of epidermal examinations obtained by us indicate rather the necrotic effect of DON, and not apoptotic. Published data indicate that in the toxic mechanism of trichothecenes, Janus-activated kinases (JAK) could induce both cell death through involvement in intracellular signal transduction in the JAK-STAT pathway. JAK, STAT1, and STAT3 are supposed to be the downstream targets for the regulation of proinflammatory response, cell proliferation, and apoptosis/necrosis induced by trichothecenes [7,40]. As published data suggests, the thickness of the skin is also correlated with its hydration [5,31]. In dogs, ultrasonography skin studies have shown that cutaneous site and hydration were correlated with skin thickness. Perhaps, a statistically significant increase in skin thickness observed in our studies, was not only result of toxic keratin damage, but also increased hydration of the skin especially superficial dermis. Superficial dermis has a loose construction, which results from the irregularly arranged collagen fibers and relatively thin elastin fibers. The deep dermis contains thick and densely arranged collagen fibers stacked parallel to the skin surface and elastin fibers that are thicker than those in the superficial dermis [31,32].

## 5. Conclusions

Skin quality is an essential indicator of welfare of farmed minks and its quality is one of the factors determining the occurrence of wounds [41]. Anatomical structure of the skin in living minks determines its durability, flexibility, and susceptibility to burst, which could cause an increased risk of infection. In mink farms in European countries, a guide to good animal welfare practice assessment protocol for mink [42] clearly state that animals should be free of injuries, skin damages, and skin wound-associated pain. This makes the skin quality one of the crucial welfare criteria of farmed minks’ good health.

The results of our studies indicate clearly that DON in minks feed causes collagen destruction in the dermis and increases skin thickness. Bentonite supplemented at the level of 0.2% restores the presence of immature collagen and increases the percentage of the elastin. However, clarification of the effects of DON and bentonite on changes in mink skin requires further research.

## Figures and Tables

**Figure 1 animals-09-01081-f001:**
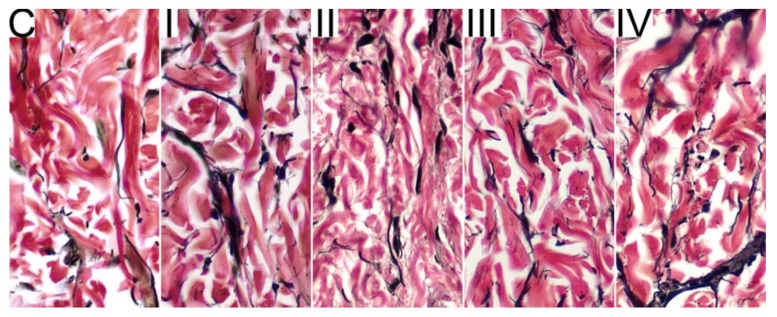
The results of elastic fibers in tissue (Verhoeff-van Gieson stain), where collagen fibers are red and elastin fibers and nuclei are black. The percentage of elastin fibers was evaluated at the magnification x400 of the microscope.

**Figure 2 animals-09-01081-f002:**
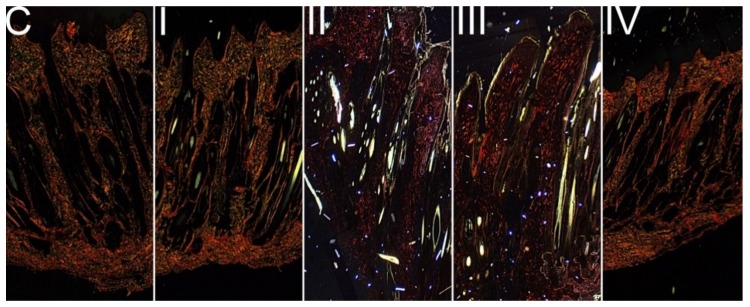
The results of collagen fibers (PSR-stained) in the connective tissue, where thinner fibers (type III or immature collagen) are green and thicker collagen fibers (type I or mature collagen) are red. The visible bright areas, shows the change of light polarization of light traveling through birefringent structures of hair shaft cortex [30]. The PSR-stained sections were evaluated in the magnification ×200.

**Table 1 animals-09-01081-t001:** The description of experimental groups. DON: deoxynivalenol (DON, vomitoxin).

Item	Control (C)	Group I	Group II	Group III	Group IV
DON contamination (mg/kg)	0	1.1	3.7	3.7	3.7
Bentonite supplementation (%)	0	0	0	0.05	0.2

**Table 2 animals-09-01081-t002:** Epidermal and dermis thickness.

Skin Layer	Control (C)	Group I	Group II	Group III	Group IV
Epidermis (µm)	17.62 ± 4.78	20.09 ± 6.78	22.55 ± 6.79 ^A,b^	23.79 ± 7.62 ^a,b^	17.25 ± 4.35 ^C,d^
Dermis (µm)	507.25 ± 79.69	1307.46 ± 459.65 ^A^	1261.39 ± 316.25 ^A^	1210.85 ± 562.61 ^A^	1016.31 ± 320.62 ^A,c,D^

Data are mean values ± SD, n = 15 in each group. ^A^ Significant difference (*p* < 0.01) compared to control group (C); ^C^ Statistically significant difference (*p* < 0.01) compared to group II; ^D^ Statistically significant difference (*p* < 0.01) compared to group III; ^a^ Statistically significant difference (*p* < 0.05) compared to control group (C); ^b^ Statistically significant difference (*p* < 0.05) compared to group I; ^c^ Statistically significant difference (*p* < 0.05) compared to group II; ^d^ Statistically significant difference (*p* < 0.05) compared to group III.

**Table 3 animals-09-01081-t003:** Elastin content and collagen to elastin ratio in the connective tissue.

Item	Control (C)	Group I	Group II	Group III	Group IV
Elastin (%)	5.90 ± 1.05	9.42 ± 2.91 ^A^	6.82 ± 1.32 ^b^	7.53 ± 1.94 ^a^	11.31 ± 1.84 ^A,b,C,D^
Collagen/elastin	10.13 ± 1.62	5.76 ± 1.62 ^A^	7.35 ± 1.73 ^A,b^	6.76 ± 1.67 ^A^	3.51 ± 0.69 ^A,B,C,D^

Data are mean values ± SD, n = 15 in each group. ^A^ Significant difference (*p* < 0.01) compared to control group (C); ^B^ Significant difference (*p* < 0.01) compared to group I; ^C^ Statistically significant difference (*p* < 0.01) compared to group II; ^D^ Statistically significant difference (*p* < 0.01) compared to group III; ^a^ Statistically significant difference (*p* < 0.05) compared to control group (C); ^b^ Statistically significant difference (*p* < 0.05) compared to group I.

**Table 4 animals-09-01081-t004:** Immature (type III) to mature (type I) collagen fibres ratio.

Control (C)	Group I	Group II	Group III	Group IV
0.163 ± 0.055	0.113 ± 0.050 ^a^	type III not observed ^1^	type III not observed ^1^	0.021 ± 0.012 ^A,B^

Data are mean values ± SD, n = 15 in each group. ^1^ Collagen type III fibers (immature) collagen were not observed in PSR-stained sections.; ^A^ Significant difference (*p* < 0.01) compared to control group (C); ^B^ Statistically significant difference (*p* < 0.01) compared to group I; ^a^ Statistically significant difference (*p* < 0.05) compared to control group (C).

**Table 5 animals-09-01081-t005:** Spearman’s rank correlation coefficients between collagen and elastin content.

Control (C)		Group I		Group II		Group III		Group IV	
Corr. ^1^	*p* ^2^	Corr.	*p*	Corr.	*p*	Corr.	*p*	Corr.	*p*
−0.782	<0.001	−0.961	<0.001	−0.678	0.005	−0.954	<0.001	−0.804	<0.001

^1^ Spearman’s rank correlation coefficient. ^2^
*p*-value (statistical significance level).

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
