# Peer review of "Evaluation of Collagen and Elastin Content in Skin of Multiparous Minks Receiving Feed Contaminated with Deoxynivalenol (DON, vomitoxin) with or without Bentonite Supplementation"

_animals, 2019, doi:10.3390/ani9121081_

Round 1

Reviewer 1 Report

The paper entitled “Evaluation of collagen and elastin content in skin of multiparous minks receiving feed contaminated with deoxynivalenol (DON, vomitoxin) with or without bentonite supplementation” by I. Taszkun et al. addresses an interesting and important subject within the important area of the studies about the effects of mycotoxins on animal health. This study are really important since mycotoxin contamination is common not only in fur framing and animal welfare should not be only a key priority in the European Union fur farming sector. Therefore, the topic fits very well within the scope of Animals journal.

The work is very interesting, applied research methods do allow to draw conclusions shown by authors. The experiment was well planned and evaluated the most important histological properties of skin tissue of farmed minks fed feed contaminated with DON. Authors found that animals exposed to DON were characterized by changed histological structure of their skin. Also the authors present strong evidence that bentonite supplementation significantly diminishes DON-induced changes in minks’ skin tissue.

However, the manuscript requires some revision before it can be recommended for publication. The authors should especially address some concerns with the presentation of the results, which are not presented well. In my opinion, the authors should consider complete rewriting of results, The first and the second paragraph of the results section presents the same results, discuss the same results, only that the second time they are supported by the results of statistical analysis. Moreover, Authors should carefully read the whole results section as some results presented in the text do not correspond with the significance levels presented in tables. For example, DON administered at a dietary dose of 1.1 mg/kg (group I) resulted in an increase in the thickness of the epidermis and dermis but the superscripts in Table 1 shows that the increase was only significant for dermis. Finally, interchangeable use of group naming (with group number, or with DON/bentonite content), makes the reading not easy for the reader.

Minor comments:

Materials and Methods

“The average euthanized multiparous mink body weight was within the range of 1477 g to 1952 g.” – Was body weight significantly different in experimental groups?

“Automated analyzes of elastic fibers were conducted on microscopic images” – How? Using specialized software?

“A color threshold instrument was used” – Again, please name the software used.

Discussion

This section is generally well-written; however, some minor changes would help emphasize the author's findings. I provide some considerations in order to additionally improve this paper.

- The effect of mycotoxins (Ochratoxin A, DON) on collagen synthesis was previously observed in studies performed cell lines, showing that mycotoxins can influence the expression of fibrous collagens, including Col3 (doi: 10.1016/j.toxlet.2018.04.014; 10.1159/000335046). Moreover, in their previous works, in which authors use the same experimental design to evaluate the effect of DON/bentonite on bone mechanical properties in minks (doi: 10.1515/jvetres-2017-0047; 10.1515/jvetres-2016-0033), authors suggest that malnutrition resulting from the presence of mycotoxicosis in the diet could impaired synthesis of Col1 collagen. As the present study confirms these results and assumptions, maybe the authors should refer to those previous works.

The welfare aspect of presented results can also be strengthened. There is a study published previously in Animals indicating that mechanical factors associated with cage design play a role in the development of skin wounds in minks (doi: 10.3390/ani8110214). Strength and flexibility of minks’ skin determines its susceptibility to occurrence of lesions, which could cause an increased risk of infection. In European countries, the Welfare assessment protocol for mink (WelFur protocol, www.sustainablefur.com), emphasize that animals should be free of injuries, especially skin damages and skin wound-associated pain, and the skin quality is welfare criteria of farmed minks good health.

Overall, despite the above remarks and suggestions, in my opinion the manuscript presents valuable data and results which will be of interest not only for mink breeders community, but also other animal feed researchers and veterinarians. Therefore, I recommend this article for publication with minor changes.

Author Response

Dear Reviewer,

Kindly refer to your points regarding our manuscript entitled “Evaluation of collagen and elastin content in skin of multiparous minks receiving feed contaminated with deoxynivalenol (DON, vomitoxin) with or without bentonite supplementation”. On your’s suggestions and recommendations, we have carefully modified the manuscript and we hope that the revised-manuscript will meet the standards of publication in journal.

Sincerely,

Iwona Taszkun

Response to Reviewer Comments

Point 1: However, the manuscript requires some revision before it can be recommended for publication. The authors should especially address some concerns with the presentation of the results, which are not presented well. In my opinion, the authors should consider complete rewriting of results, The first and the second paragraph of the results section presents the same results, discuss the same results, only that the second time they are supported by the results of statistical analysis. Moreover, Authors should carefully read the whole results section as some results presented in the text do not correspond with the significance levels presented in tables. For example, DON administered at a dietary dose of 1.1 mg/kg (group I) resulted in an increase in the thickness of the epidermis and dermis but the superscripts in Table 1 shows that the increase was only significant for dermis. Finally, interchangeable use of group naming (with group number, or with DON/bentonite content), makes the reading not easy for the reader.

Thank you for pointing out our mistakes. We are sorry for them. We decided to completely rewrite the results section to make it more clear and remove any inaccuracies and repetitions in the text (L172-220).

Point 2:  “The average euthanized multiparous mink body weight was within the range of 1477 g to 1952 g.” – Was body weight significantly different in experimental groups?

Thank you for pointing this out. The body weight was significantly reduced in the group fed wheat contaminated with DON at the dose of 3.3 mg/kg (group II). We added this information to the manuscript (L115-117).

Point 3: “Automated analyzes of elastic fibers were conducted on microscopic images” – How? Using specialized software? , “A color threshold instrument was used” – Again, please name the software used.

All analyses were performed using appropriate tools in ImageJ software. We added this information to the manuscript (L160).

Point 4: The effect of mycotoxins (Ochratoxin A, DON) on collagen synthesis was previously observed in studies performed cell lines, showing that mycotoxins can influence the expression of fibrous collagens, including Col3 (doi: 10.1016/j.toxlet.2018.04.014; 10.1159/000335046). Moreover, in their previous works, in which authors use the same experimental design to evaluate the effect of DON/bentonite on bone mechanical properties in minks (doi: 10.1515/jvetres-2017-0047; 10.1515/jvetres-2016-0033), authors suggest that malnutrition resulting from the presence of mycotoxicosis in the diet could impaired synthesis of Col1 collagen. As the present study confirms these results and assumptions, maybe the authors should refer to those previous works.

Thank you for this comment. The previously reported information about the effects of mycotoxin on collagen synthesis have been discussed in detail. For this reason we have decided to reorganize the entire paragraph in the discussion (L161-276).

Point 5: The welfare aspect of presented results can also be strengthened. There is a study published previously in Animals indicating that mechanical factors associated with cage design play a role in the development of skin wounds in minks (doi: 10.3390/ani8110214). Strength and flexibility of minks’ skin determines its susceptibility to occurrence of lesions, which could cause an increased risk of infection. In European countries, the Welfare assessment protocol for mink (WelFur protocol, www.sustainablefur.com), emphasize that animals should be free of injuries, especially skin damages and skin wound-associated pain, and the skin quality is welfare criteria of farmed minks good health.

Thank you for your nice suggestion. We added a whole new paragraph discussing the welfare aspects of farmed minks (L301-L307).

We also corrected other minor errors and typos noticed at work. They were also marked red.

Reviewer 2 Report

This is an interesting study examining the effects of DON, administrated with or without bentonite supplementation on the histological structure of the skin in farmed minks. This is an important topic and the manuscript is generally well-written; however, some changes would help clarify the authors’ findings, such as significant increase in dermis thickness in DON-exposed groups and the lack of immature collagen fibres in two groups exposed to higher dose of DON.

However, some ameliorations are needed. The list of points of concerns is below:

Abstract:

- Please change “supplemented with DON” to “exposed to DON”

Introduction:

- “skin size”? Please clarify.

- Please correct “deoxynivalenonol” to “deoxynivalenol” (also in simple summary).

- “dose of DON over 2 mg/kg feed is considered to be the LD50 toxic dose” – Is this correct? For what animal? Please verify, especially as in the next part of this paragraph authors state that “Ruminants and poultry tolerate up to 20 mg/kg of DON in feed”.

- “It is known that toxic effect of DON consist in inhibiting protein synthesis” – it was also shown by Lu et al. (Cell Tissues Organs, 2012, 196, 241-250) in in vivo study on cartilage chondrocytes that DON decrease type II collagen synthesis, while type X collagen was increased in response to DON. As the present study also analyses effects of DON on collagen, please consider adding this reference (here or in the discussion).

- “One of the protective methods against the adverse effect of DON on animal health is to use as a feed supplement to allow toxins absorption, such as bentonite”. Bentonite adsorption efficacy on DON mycotoxin depends on bentonite type (smectite content). Authors may add there the information that sodium bentonite, supplemented at the dose of  1%, was found to reduce fungus number in farmed minks’ feed (Polish J  Environ Stud, 2011, 20, 1103-1106) without adversely affecting animal health, as indicated by blood parameters (Slov Vet Res, 2015, 52, 165-171).

Methods:

- How was feed contaminated with DON?

- On what basis the used doses of bentonite Mycofix bentonite/dioctahedral montmorillonite were selected?

DON has a negative effect on growth. Did the minks body weight differ between groups?

- Why were skin samples taken from this area?

- “A color threshold instrument was used to select a range of dark brown to black elastic fibers from the cross section of the dermis images” – which graphical software was used?

- How the mature/immature collagen content was determined?

Results:

- The results section needs improvements, as the presentation of the results is difficult to understand and sometimes the text does not match the data in tables. It is highly recommended to read the entire section carefully and thoroughly correct. Moreover, in the section where statistical analysis were described, authors state that only the probability level of P < 0.05 and P < 0.01 were considered, which is consistent with the results presented in tables. However, in the text, the significance level is sometimes marked at the probability level of P < 0.001.

- Figure 2 – could you explain the origin of bright areas visible especially in groups II and III. Did it affect the analyses?

Discussion:

Overall the discussion is well written and comprehensive, focused on the topic, with good reference to the other. I may only suggest highlighting the animal welfare aspect of this study.

 I suggest citing other publications related to the use of bentonite as an additive to mink feed, for example: The effect of sodium bentonite supplementation in the diet of mink (Neovison vison) on the microbiological quality of feed and animal health parameters  or  Effect of dietary sodium bentonite supplement on microbial contamination of mink feed.

Concluding, concept of this study is very interesting, and the result is worth to publish in Animals after revision.

Author Response

Dear Reviwer,

We would like to thank you and the reviewer for taking the time to review our manuscript and to give valuable comments. We have made corrections and clarifications in the manuscript according to the reviewer’s comments.

Sincerely,

Iwona Taszkun

The changes are summarized below

Point 1: Please change “supplemented with DON” to “exposed to DON”

Corrected.

Point 2: “skin size”? Please clarify.

We have rewritten this sentence add have additional reference [1].

Point 3: - Please correct “deoxynivalenonol” to “deoxynivalenol” (also in simple summary).

Sorry for this typo error. It has been corrected.

Point 4: - “dose of DON over 2 mg/kg feed is considered to be the LD50 toxic dose” – Is this correct? For what animal? Please verify, especially as in the next part of this paragraph authors state that “Ruminants and poultry tolerate up to 20 mg/kg of DON in feed”.

Corrected to 2mg/kg of body weight. We also have rewritten this part to make it more definite (L67-L74).

Point 5: “It is known that toxic effect of DON consist in inhibiting protein synthesis” – it was also shown by Lu et al. (Cell Tissues Organs, 2012, 196, 241-250) in in vivo study on cartilage chondrocytes that DON decrease type II collagen synthesis, while type X collagen was increased in response to DON. As the present study also analyses effects of DON on collagen, please consider adding this reference (here or in the discussion).

Thanks for your suggestions. We have decided to separate the paragraph in the discussion where we discuss these results (L161-276).

Point 6: - “One of the protective methods against the adverse effect of DON on animal health is to use as a feed supplement to allow toxins absorption, such as bentonite”. Bentonite adsorption efficacy on DON mycotoxin depends on bentonite type (smectite content). Authors may add there the information that sodium bentonite, supplemented at the dose of  1%, was found to reduce fungus number in farmed minks’ feed (Polish J  Environ Stud, 2011, 20, 1103-1106) without adversely affecting animal health, as indicated by blood parameters (Slov Vet Res, 2015, 52, 165-171).

Thanks for your valuable comments, which we included in the introduction of our revised text (L78-81), ref. [17, 19, 20].

Point 7: - How was feed contaminated with DON?

Wheat free of mycotoxins was mixed with wheat naturally contaminated with DON which level las determined by HPLC. The DON content in given feed was also verified by HPLC analyses. We provided the more clearly details in the revised manuscript (L107-111).

Point 8: - On what basis the used doses of bentonite Mycofix bentonite/dioctahedral montmorillonite were selected?

The doses were selected on the basis of EFSA recommendations and working levels recommended by the producer [18, 23]. L(110-111)

Point 9: DON has a negative effect on growth. Did the minks body weight differ between groups?

Thank you for pointing this out. The body weight was significantly reduced in the group fed wheat contaminated with DON at the dose of 3.3 mg/kg (group II). We added this information to the manuscript (L115-117).

Point 10: - Why were skin samples taken from this area?

Study was a conducted on a commercial mink farm. The samples were collected before pelt harvesting, as during that process skin is being stretched and its internal structure changes. Additionally, skin endurance at this area of the limb has technological significance, as it is place where the incisions during skin harvesting are made.

Point 11: - “A color threshold instrument was used to select a range of dark brown to black elastic fibers from the cross section of the dermis images” – which graphical software was used?

All analyses were performed using appropriate tools in ImageJ software. We have added this information to the manuscript (L160).

Point 12: - How the mature/immature collagen content was determined?

All analyses were performed using ImageJ software, as described in [26]. However, we decided to add some description, due to hair birefringence (see answer to Point 14) (L153-157).

Point 13: - The results section needs improvements, as the presentation of the results is difficult to understand and sometimes the text does not match the data in tables. It is highly recommended to read the entire section carefully and thoroughly correct. Moreover, in the section where statistical analysis were described, authors state that only the probability level of P < 0.05 and P < 0.01 were considered, which is consistent with the results presented in tables. However, in the text, the significance level is sometimes marked at the probability level of P < 0.001.

Thank you for pointing out our mistakes. We decided to completely rewrite the results section to make it more clear and remove any inaccuracies and repetitions in the text (L172-220).

Point 14: - Figure 2 – could you explain the origin of bright areas visible especially in groups II and III. Did it affect the analyses?

The cortex of the shaft of the hair is birefringent. Light traversing through a hair will experience birefringence, which results in a change in polarization, and thus areas along the hair axis are seen as bright areas. As added in materials and methods, only section without visible fur hair were analyzed. An explanation with the appropriate reference [30] has been given both in the material and methods section (L156-157) and the Figure 2 caption (L225-226).

Point 15: I suggest citing other publications related to the use of bentonite as an additive to mink feed, for example: The effect of sodium bentonite supplementation in the diet of mink (Neovison vison) on the microbiological quality of feed and animal health parameters  or  Effect of dietary sodium bentonite supplement on microbial contamination of mink feed.

Thank you for pointing out these interesting works. We added the reference to those and other work related to the use of bentonite in mink farms both, in introduction and discussion (see answer to Point 6), ref [19, 20, 39].

We also corrected other minor errors and typos noticed at work. They were also marked red.
